# Low Voltage High-Energy α-Particle Detectors by GaN-on-GaN Schottky Diodes with Record-High Charge Collection Efficiency

**DOI:** 10.3390/s19235107

**Published:** 2019-11-21

**Authors:** Abhinay Sandupatla, Subramaniam Arulkumaran, Kumud Ranjan, Geok Ing Ng, Peter P. Murmu, John Kennedy, Shugo Nitta, Yoshio Honda, Manato Deki, Hiroshi Amano

**Affiliations:** 1School of Electrical and Electronics Engineering, Nanyang Technological University, Singapore 639798, Singapore; 2Temasek Laboratories @ NTU, Research Techno Plaza, 50 Nanyang Drive, Singapore 639798, SingaporeKRanjan@ntu.edu.sg (K.R.); 3Center for Integrated Research of Future Electronics (CIRFE), IMaSS, Nagoya University, Nagoya 464-8603, Japanhonda@nuee.nagoya-u.ac.jp (Y.H.); deki@nuee.nagoya-u.ac.jp (M.D.); amano@nuee.nagoya-u.ac.jp (H.A.); 4National Isotope Center, GNS Science, Lower Hutt 5010, New Zealand; P.Murmu@gns.cri.nz (P.P.M.); J.Kennedy@gns.cri.nz (J.K.)

**Keywords:** high-energy α-particle detection, low voltage, thick depletion width detectors

## Abstract

A low voltage (−20 V) operating high-energy (5.48 MeV) α-particle detector with a high charge collection efficiency (CCE) of approximately 65% was observed from the compensated (7.7 × 10^14^ /cm^3^) metalorganic vapor phase epitaxy (MOVPE) grown 15 µm thick drift layer gallium nitride (GaN) Schottky diodes on free-standing n+-GaN substrate. The observed CCE was 30% higher than the bulk GaN (400 µm)-based Schottky barrier diodes (SBD) at −20 V. This is the first report of α–particle detection at 5.48 MeV with a high CCE at −20 V operation. In addition, the detectors also exhibited a three-times smaller variation in CCE (0.12 %/V) with a change in bias conditions from −120 V to −20 V. The dramatic reduction in CCE variation with voltage and improved CCE was a result of the reduced charge carrier density (CCD) due to the compensation by Mg in the grown drift layer (DL), which resulted in the increased depletion width (DW) of the fabricated GaN SBDs. The SBDs also reached a CCE of approximately 96.7% at −300 V.

## 1. Introduction

The measurement of the energy spectrum of charged particles plays an important role in studying the fusion reactions of nuclear reactors and in particle physics research conducted at facilities like the Large Hadron Collider. Gallium Arsenide (GaAs) has been the primary contender as an alternative to Si-based detectors. However, due to its low displacement energy (E_d_) of 10 eV [1], the lifetime of the GaAs detector is limited. Low E_d_ results in the easier displacement of Ga and As atoms from their respective crystal lattice, thereby weakening their lattice structure. Gallium Nitride (GaN) was considered as an alternative in high-energy charged particle detection due to its higher E_d_ (20 eV) [1]. The high E_d_ of GaN would improve the radiation tolerance of the GaN detector, as reported by B.D. Weaver et al., by comparing the radiation damage on GaN and GaAs [2]. The report also determined that GaN could withstand twice the dosage in comparison to GaAs. GaN also has a larger bandgap of 3.4 eV [3] in comparison to GaAs (1.42 eV) and Si (1.12 eV), which enables GaN detectors to operate at higher temperatures. The superior material characteristics of GaN have encouraged multiple research groups in exploring applications of GaN devices in radiation detection. GaN devices have performed exceedingly well as α-particle detectors [4,5,6,7,8,9,10,11,12,13] and neutron detectors [14,15,16,17]. The structure of these GaN detectors can be classified into three types, namely double Schottky contact (DSC) structure, mesa structure and sandwich structure.

The first ever GaN alpha particle detector was realized by Vaitkus et al. [4] with the DSC structure. The detector had a 2 μm GaN epi-layer, which resulted in the detection of 410 keV alpha particles with a 92% charge collection efficiency (CCE). Similarly, other research groups have successfully implemented different types of mesa structures and achieved the higher CCE of 100% due to a lower trapping impurity concentration. Compared to DSC structures and mesa structures, sandwich structures have the potential to generate the thickest depletion widths (DWs), which enables the detection of higher energies. The development of the sandwich structures of GaN detectors was primarily limited by the unavailability of free-standing substrates. With the improvement in the GaN growth technologies, researchers have been able to produce high-quality free-standing GaN substrates and epitaxial films. This has led many research groups to explore the sandwich structure for GaN-based radiation sensors [4,5,6,8]. GaN detectors with thin epitaxial films detect only fractions of the typical energies emitted by actinides (4 MeV to 6 MeV), such as U-235 (4.268 MeV), ^241^Am (5.48 MeV) and Pu (4.67 MeV). While different thicknesses of epitaxial drift layers (DLs) (2 μm to 12 μm) were tested, they could only detect energies in the range of 0.5 MeV [4] to 4.5 MeV [5], which is on the lower end of detection requirements. To detect higher energies, researchers increased the DWs of the detector by fabricating them on bulk GaN substrates. These detectors generate a 27 μm DW at very high voltages (−550 V) to detect high energies (5.48 MeV) [6]. The high voltage required to generate a thick DW in bulk GaN-based detectors increases both the complexity and size of the detector, thus severely affecting its portability. In this work, for the first time we design and fabricate GaN-on-GaN Schottky barrier diodes with compensated metalorganic vapor phase epitaxy (MOVPE) grown GaN DL to detect 5.48 MeV α-particles (^241^Am source), even at −20 V.

## 2. Design of α-Particle Detector

To design an α-particle detector that works in low-bias conditions and detects high α-particle energies, we employed GaN-on-GaN Schottky barrier diodes (SBDs) with a sandwich structure (see Figure 1). The thin epitaxial GaN layer was very lowly doped, resulting in a thick DW, while the highly doped substrate gave mechanical strength to the detector and formed a good ohmic contact, which helped to reduce the biasing voltage needed.

The thickness of the DL plays an important role in determining the maximum thickness of DW, thereby the maximum energy that can be detected. The DL required to detect 5.48 MeV emitted by ^241^Am was calculated through Stopping and Range of Ions in Matter (SRIM) simulations by considering GaN density (6.1 g/cm^3^) and α-particle mass (4.003 amu). Figure 2 shows the SRIM-calculated range of the thickness required in GaN to absorb α-particles with energies between 10 keV and 6 MeV. From this graph, we found that we needed a 14.54 μm DW to detect α-particles with 5.48 MeV. Hence, SBDs with a DL of minimum 15 µm were required for 5.48 MeV α-particle detection. 

The 15 μm thick GaN DL was grown by MOVPE at 1080 °C (100 kPa) on hydride vapor phase epitaxy (HVPE)-grown free-standing GaN substrates with a charge carrier density (CCD) of 1 × 10^18^/cm^3^. The growth rate of 3.5 μm/h was maintained during the growth. The grown DL exhibited low CCD, which was measured using Van der Pauw, Hall and secondary ion mass spectrometry (SIMS) analyses (See Figure 3 and Table 1). From SIMS analysis, the concentrations of different elements present were extracted. While the measured values of O were below the detection limit (3 × 10^15^ /cm^3^), all C, Si, Mg and Fe were present. For the calculation of CCD, we list the extracted concentrations of Si and Mg in Table 1. Though the calculated CCD from SIMS was nominally negligible, the calculated values from SIMS were in a similar range of CCD measured by Hall measurements. Panchromatic cathodoluminescence (CL) measurements were also performed to count the threading dislocation density (TDD), which was an average of 2.65 × 10^6^/cm^2^ on the MOVPE-grown GaN DL.

## 3. Detector Fabrication and Measurement Setup

### 3.1. Detector Fabrication

The fabrication of SBDs started with a thorough cleaning of the GaN-on-GaN wafer with piranha solution and organic cleaning (acetone and isopropanol) and dipping in buffered oxide etchant (BOE) for two minutes to ensure the formation of an excellent metal-semiconductor interface [18]. After the surface preparation, the ohmic contact was formed by depositing Ti/Al/Ni/Au (20/120/40/50 nm) on the N-face (backside) of the wafer, followed by rapid thermal annealing at 775 °C for 30 s in N_2_ ambience. The selection of Ti was to form a low-resistance contact as Ti helps generate large amounts of N-vacancies after annealing [18], which increases CCD and promotes tunneling. The second element Al was used to absorb excessive Ti material [19], while Ni was used as a barrier metal, which confines the downward diffusion of the fourth layer (Au) [20]. The top layer of Au was required to protect layers below from oxidization [21]. Multiple SBDs of different sizes were then fabricated by depositing Ni/Au (50/1000 nm) on the Ga-face of the wafer. Ni was selected as the first layer due to the difference in work functions of Ni (5.04 eV) and GaN (4.2 eV) [22], which helps form Schottky contact. After the formation of SBDs, electrical characterization was performed, followed by the dicing of the wafer into individual SBDs. Each SBD was then mounted on a dual in-line package (DIP) by connecting the ohmic contact of the SBD and the ground of the DIP with conductive Ag paste. The Schottky contact was wire bonded using 20 μm diameter Au wires (see Figure 4).

### 3.2. α-Particle Measurement Setup

^241^Am was used as a source for the generation of 5.48 MeV α-particles with an active area of 7 mm^2^ placed at 8 mm from the detector (as shown in Figure 5). The α-particle source had radionuclides deposited onto a stainless-steel disc of 16 mm diameter, which was held in place by a plastic holder. The GaN SBD detector was connected to a pre-amplifier, amplifier and signal-processing circuit to detect the change in the current flowing through the SBDs due to the interaction with an α-particle. A Si surface barrier detector from ORTEC was used as a reference, along with an ORTEC-671 amplifier for energy calibration.

## 4. Results and Discussion

### 4.1. Current–Voltage (I–V) Characteristics 

The *I–V* characteristics were measured using a B1505A power device analyzer at room temperature. Mg-compensation of the GaN DL helped to reduce the CCD by two orders of magnitude, from 4.6 × 10^16^/cm^3^ to 7.7 × 10^14^/cm^3^. The huge reduction in CCD resulted in the increase of the breakdown voltage by approximately three times, from 462 V to 1480 V, and the reduction of the reverse leakage current by approximately three orders (see Figure 6). A very low reverse leakage current of 3 pA at −20 V bias is low enough for event-by-event counting to acquire α-particle energy spectra. The SBD also exhibited an average ideality factor and Schottky barrier height of 1.03 and 0.79 eV, respectively [23,24,25]. The near-unity ideality factor signified an excellent metal-semiconductor interface at the Schottky-semiconductor contact [26]. Similarly, the extracted barrier height of 0.79 eV was similar to other reported Ni-based Schottky contacts [27,28].

### 4.2. Capacitance–Voltage (C–V) Characteristics

C–V measurements were also performed to extract the DW of the GaN-on-GaN SBDs. No significant variation in capacitance value was observed for a voltage range of −20 V to 5 V (see Figure 7), which signifies the complete depletion of the DL [10,29].

DW can be extracted from the C–V characteristics using Equation (1):(1)$$C=\;\varepsilon_{0}\varepsilon_{r}\left(A/DW\right)$$

A uniform DW of approximately 15 μm was measured at all voltages (−20 V to 5 V), which implies the complete DL is depleted even at 0 V.

### 4.3. Detection of α-Particle Spectra

The performance of an α-particle detector is primarily defined by its CCE. CCE is the ratio of detected energy and incident energy, which is dependent on the DW of the detector. The acquired data was calibrated using a standard Si detector as a reference. In this process, the fabricated GaN detector and the reference Si detector were connected to the same measurement setup separately, without changing the settings. The final detected energy is described by the following equation [6]:E= E_0_ + W_GaN_/W_Si_ × k × Channel(2)
where E is the absorbed energy; E_0_ represents energy loss at the metal interface, which was estimated based on the Transport of Ions in Matter (TRIM) simulation to be 183 keV for an Au/Ni (1000/50 nm) contact; k is a calibration factor of the reference Si detector; W_GaN_ (8.9 eV) [30] and W_Si_ (3.6 eV) are the energies required to generate an electron-hole pair in GaN and Si, respectively.

#### 4.3.1. Variation in α-Particle Spectra-Air vs. Vacuum

Figure 8 shows the comparison of the energy spectrum of GaN detectors biased at −100 V measured in a vacuum and in air. About a 7% reduction in CCE was observed when the detectors were measured in air. When an α-particle passes through the vacuum, all its energy is transferred to the detector, resulting in high-energy detection. When α-particles traverse through air, scattering results in loss of energy. This loss in α-particle energy results in a lower CCE. For practical applications, portability and cost are important factors and the need for a vacuum restricts portability and also increases the cost of the system. While the presence of air results in a 7% drop in CCE, compensated SBDs could still be considered for practical applications.

#### 4.3.2. Low Voltage α-Particle Detection 

The α-particle energy spectra obtained under low-bias conditions (−20 to −80V) are shown in Figure 9a. It can be observed that with the decrease in applied bias, the detected energy also decreases. The decrease in detected energy is due to the reduction in DW at lower bias conditions. Reduced DW decreases the path length of the α-particle inside DW, thereby limiting the ability to detect high-energy α-particles. To check detector performance uniformity, four detectors were tested under similar conditions and only a small variation of approximately 2% was observed in measured CCE. Figure 9b shows the variation of CCE with the voltage of our detectors compared with other published bulk GaN detectors (sandwich structures). Our detectors exhibited very low variation in CCE (7%) with a change in voltage (−20 V to −80 V) when compared to the variation shown by a 450 μm thick bulk GaN detector (32.7%) [6] and a 500 μm thick bulk GaN detector (58.1%) [7]. The low variation in CCE is mainly due to the presence of a thick DW even at low-bias conditions, which was achieved by reducing CCD by the compensation of DL.

#### 4.3.3. High Voltage α-Particle Detection

Our detectors were also biased at higher voltages to obtain a thicker DW leading to a higher CCE, similar in the range reported by others using bulk GaN detectors [6,7]. Our detector performance improved from 72% at −80 V to 96.7% at −300 V (see Figure 10a), which is the lowest reported voltage at which 5.48 MeV α-particle was successfully detected. The high-voltage performance of these detectors was compared with other published α-particle detectors. Of the many published reports, only Q. Xu et al. reported a high CCE of 100% while detecting a 5.48 MeV α-particle. However, their detectors need to be biased up to −550 V, which is 250 V higher than our detectors (see Figure 10b).

The detector’s energy resolution was extracted to be 71 keV from the full wave at half maximum (FWHM) of the α-particle spectra measured at −100 V. The measured energy resolution is 30% better than other reported bulk GaN-based detectors (121 keV) [6]. The improved energy resolution was due to the reduction in the straggling (statistical distribution of energy losses) of α-particles by placing the detector normal to the source. SRIM simulations were also performed to study the effect of the incident angle on energy resolution [27] (see Figure 11). An increase in the angle between the incident α-particle and the surface of the detector increases the FWHM of the detected spectral energy from 97.8 keV to 163.8 keV. This increase in FWHM is due to the increase in straggling when the source is placed at an angle to the detector, resulting in a reduction of straggling.

### 4.4. Benchmarking

Figure 12 compares the low voltage performance of GaN-on-GaN SBD detectors with the state-of-the-art α-particle detectors reported so far, as a function of detected energies. About 30% higher CCE was observed in the compensated 15 µm thick GaN DL-based α-particle detectors at −20 V. In addition, our detectors also exhibited 96.7% CCE at −300 V, which is 250 V lower than the published literature. These promising results pave the way to achieve high CCE, low operating voltage and portable α-particle detectors.

## 5. Conclusions

In conclusion, we demonstrated a low-voltage (−20 V) operating 5.48 MeV α-particle detector with a record-high CCE of 65% using 15 µm thick compensated MOVPE-grown GaN DL on HVPE-grown bulk n+-GaN substrate. The measured CCE was 30% higher than the previously reported values at −20 V. The detectors also exhibited a high CCE of 96.7% at −300 V and the spectral resolution of 71 keV, which was 250 V lower and 30% better than the previously reported values. The improved performance in α-particle detection was due to the formation of a thicker DW, even at low voltages. The demonstrated vertical GaN-on-GaN SBD with compensated DL for portable α-particle detectors presented great potential to work even at low voltages.

## Figures and Tables

**Figure 1 sensors-19-05107-f001:**
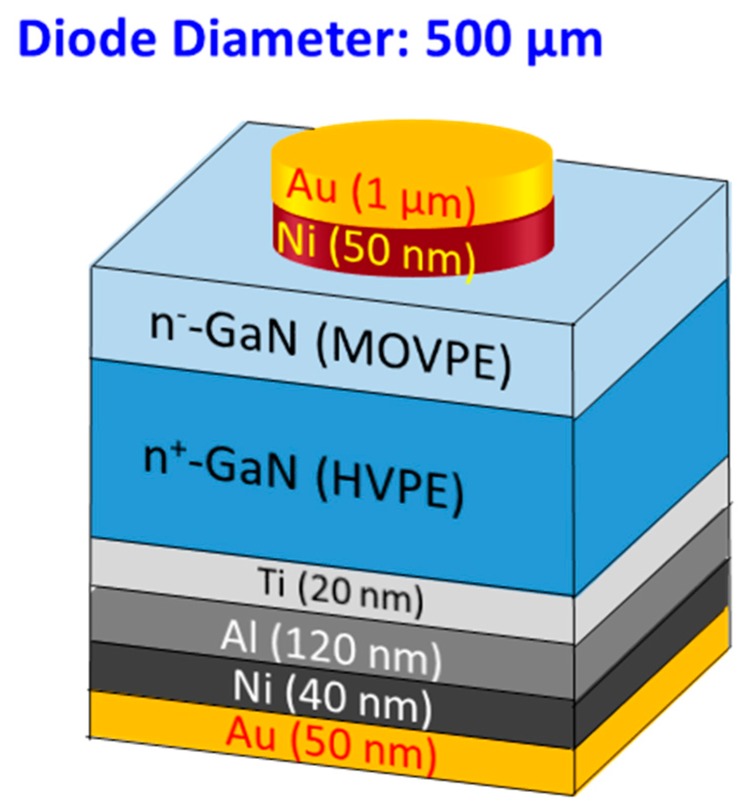
Gallium nitride-on-gallium nitride (GaN-on-GaN) Schottky barrier diodes (SBD) with a sandwich structure for alpha particle detection.

**Figure 2 sensors-19-05107-f002:**
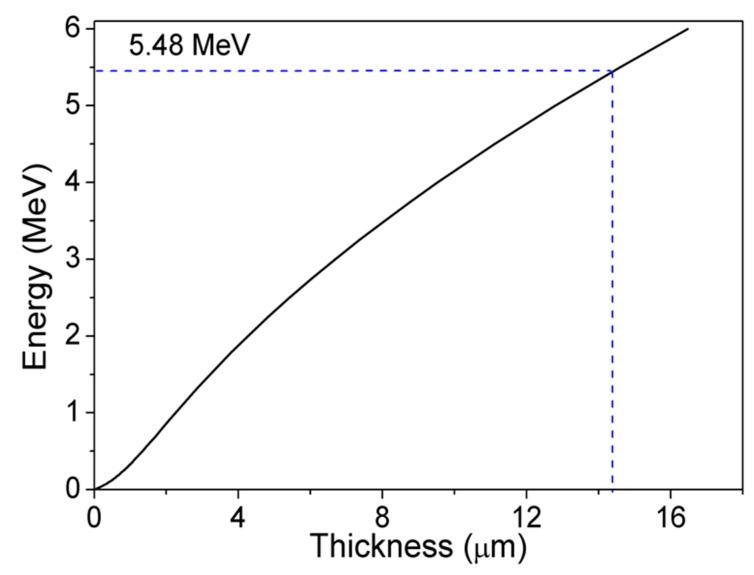
α-particle (^241^Am, 5.48 MeV) range in GaN calculated by Stopping and Range of Ions in Matter (SRIM).

**Figure 3 sensors-19-05107-f003:**
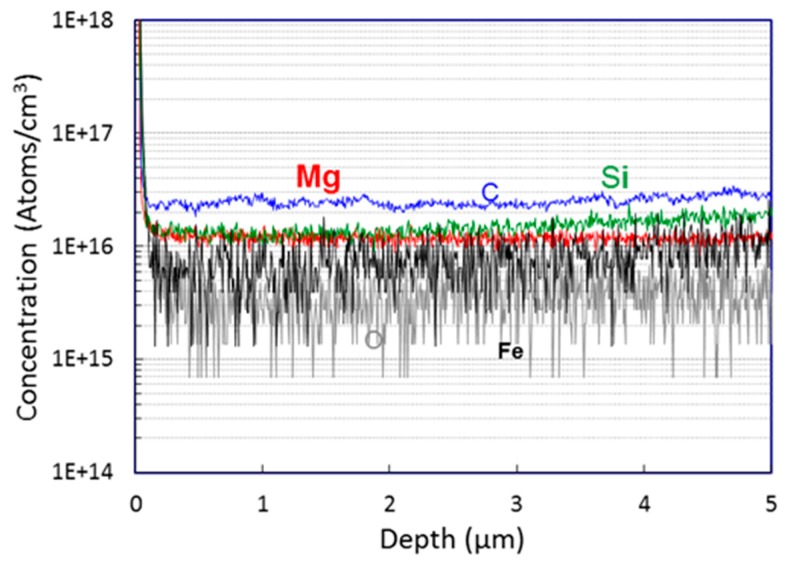
Elemental concentration extracted through secondary ion mass spectrometry (SIMS) at different depths of the drift layer (DL).

**Figure 4 sensors-19-05107-f004:**
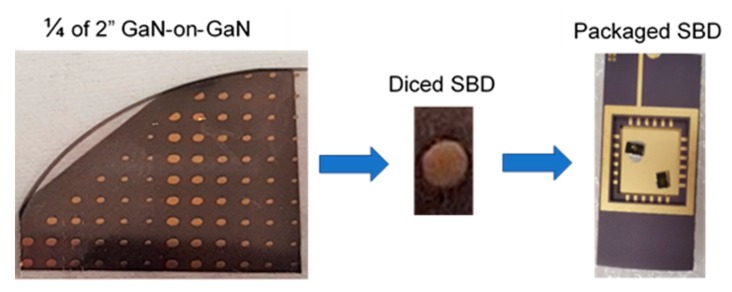
Three stages of sample preparation.

**Figure 5 sensors-19-05107-f005:**
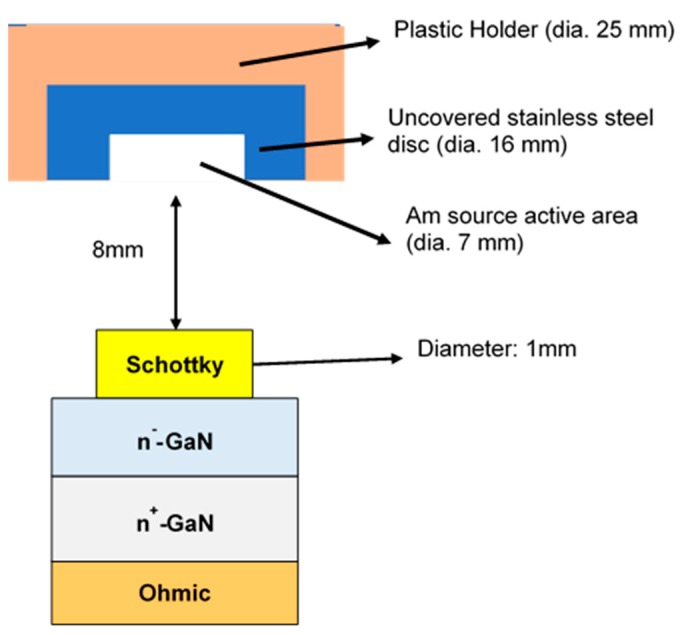
Schematic drawing of Source-Detector measurement setup (not to scale).

**Figure 6 sensors-19-05107-f006:**
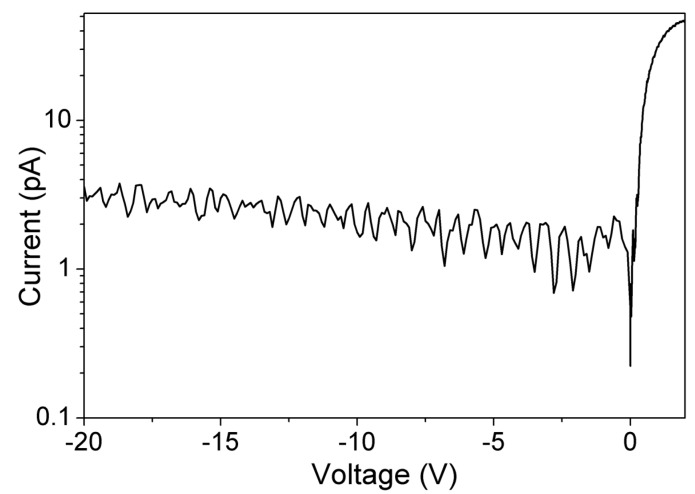
Room temperature *I–V* characteristics of 1 mm diameter GaN SBDs with 15 µm thick compensated DL.

**Figure 7 sensors-19-05107-f007:**
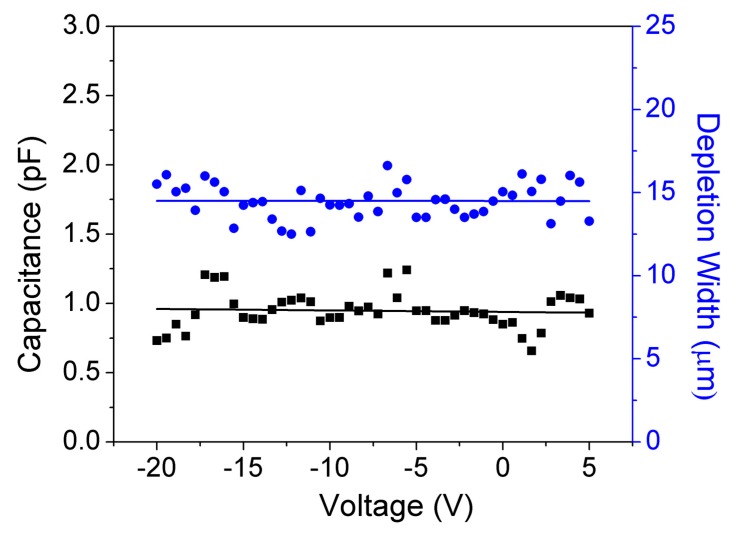
Variation of capacitance and depletion width (DW) with voltage of 0.5 mm diameter GaN SBDs with 15 µm thick Mg-compensated DL.

**Figure 8 sensors-19-05107-f008:**
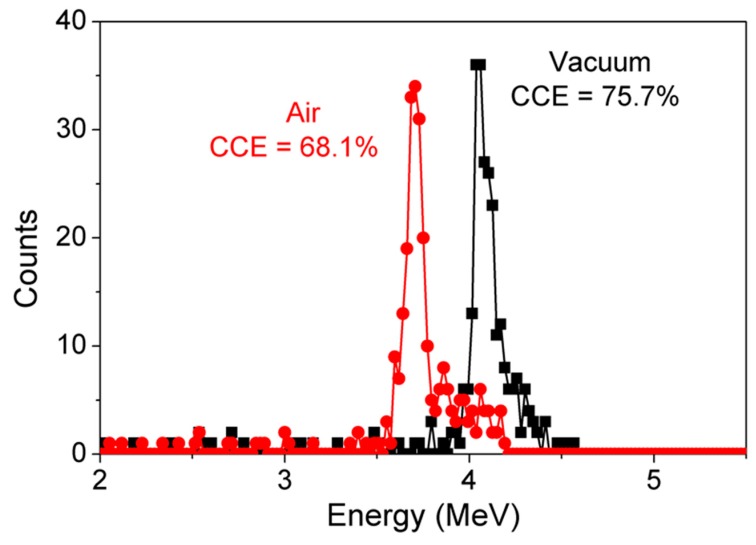
Acquired α-particle energy spectra of GaN SBDs at −100 V under air and in a vacuum.

**Figure 9 sensors-19-05107-f009:**
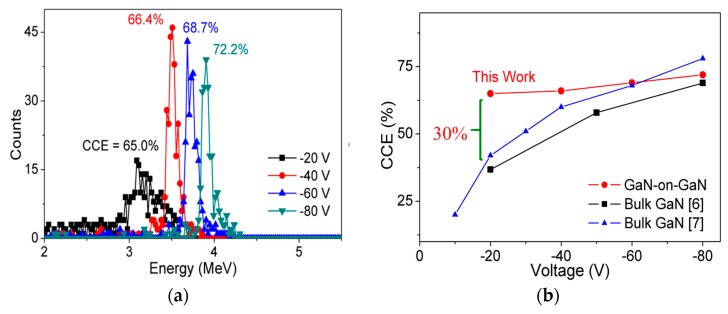
(**a**) Acquired α-particle spectra of GaN SBDs for different applied voltages (−20 V to −80 V) and (**b**) Comparison of measured charge collection efficiency (CCE) of SBDs vs. applied voltages (−20 V to −80 V) with state-of-the-art reported values.

**Figure 10 sensors-19-05107-f010:**
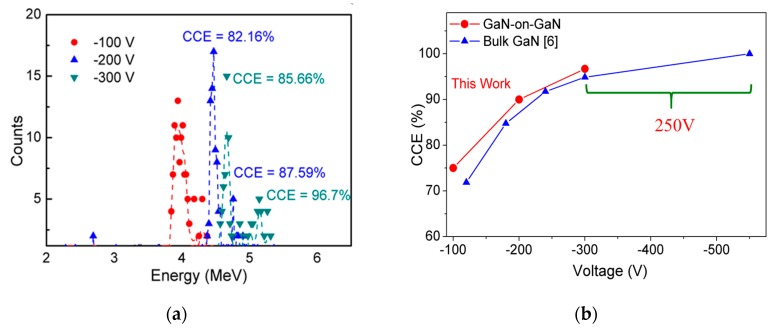
(**a**) Acquired α-particle spectra of GaN SBDs for different applied voltages (−100 V to −300 V) and (**b**) Comparison of measured CCE of SBDs vs. applied voltages (−100 V to −550 V) with state-of-the-art reported values.

**Figure 11 sensors-19-05107-f011:**
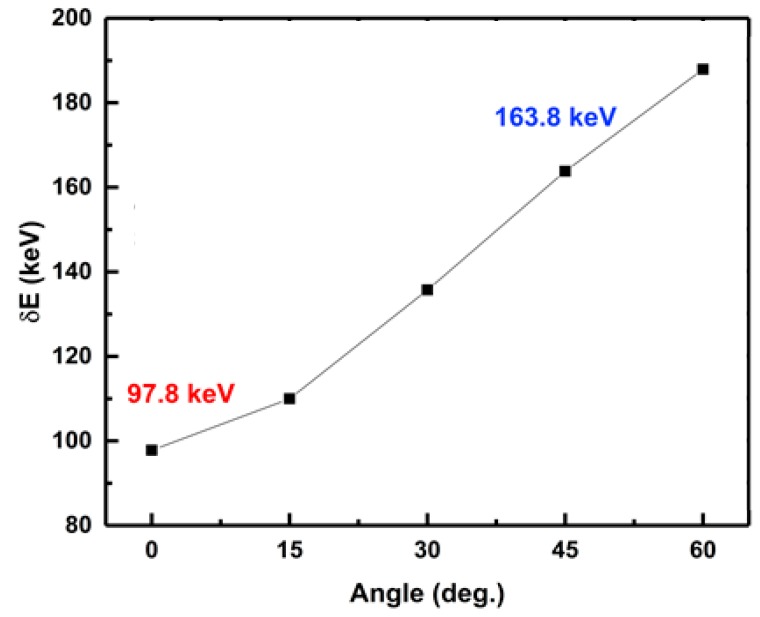
Variation of energy resolution with a change of the incident angle.

**Figure 12 sensors-19-05107-f012:**
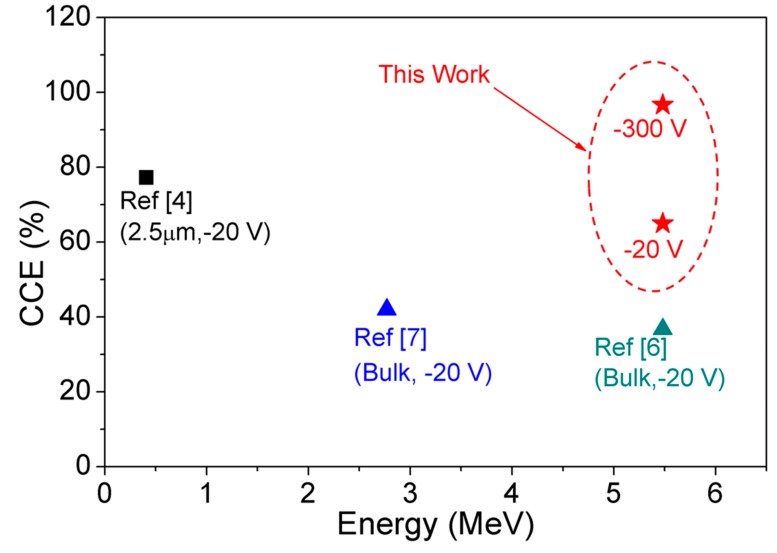
Benchmarking of extracted CCE of our detectors with epitaxial-grown GaN detectors (squares) and bulk GaN detectors (triangles) at low voltages.

**Table 1 sensors-19-05107-t001:** Concentration of Si and Mg measured through SIMS with the corresponding CCD extracted from SIMS and Hall measurements.

Si (N_D_) (/cm^3^)	Mg (N_A_) (/cm^3^)	CCD = N_D_ − N_A_ (/cm^3^)	
		SIMS	Hall
154.79 × 10^14^	147.09 × 10^14^	7.7 × 10^14^	7.5 × 10^14^

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
