# Peer review of "Low Voltage High-Energy α-Particle Detectors by GaN-on-GaN Schottky Diodes with Record-High Charge Collection Efficiency"

_sensors, 2019, doi:10.3390/s19235107_

Round 1
Reviewer 1 Report
This is a well written paper on characterization of a diode as alpha particle. I have some minor comments to improve the paper for reader who are familiar with radiation detection as photo such as gamma rays.
Please describe the displacement t energy.
In Figure 2 why there is just 14 um require to detect alpha particle.
Reviewer 2 Report
The work in the manuscript focus a specific low-voltage a detectors based on Schottky GaN structures. The topic is interesting and currently under research. The results obtained by the authors are interesting and can, eventually open new perspectives in this field of research. However, this specific kind of devices for a detectors is not new and even the “sandwich” structure has been extensively studied. The manuscript has no critical issues but in the current form, is not easy to find a real prime novelty. Therefore, and besides the interesting results, it is my opinion that the authors should improve some specific issues in the manuscript in order to be suitable for publication in Sensors. In particular, authors should consider the following points:
a) Using the SRIM (acronyms never defined) is a normal procedure but the authors should explain the considerations made (and/or approximations) in establish the conditions for simulations;
b) Almost the same for the estimated energy loss (E0) by TRIM. Was made by the authors?
c) Have the authors any possibility to estimate the responsivity and sensitivity (or even quantum efficiency), and compared with reported in the literature?
d) The data comparing response under air or vacuum conditions is interesting. But how compares with results in literature, i.e. the usual loss when detection are made under vacuum (and not the absolute values);
e) The results obtained in low voltage detection are clearly good. How is the reproduction of such results, i.e. several samples and using the same device several times?
f) Finally, and because this idea is nor really new, a more deep discussion regarding device structure and its material characterization and the results obtained is needed in order to really enhance the novelty of the work.
Reviewer 3 Report
The authors realized a sandwich structure Schottky diode on a MOVPE grown low doing low threading dislocation density n-type GaN on an n+-GaN substrate for alpha particle detection. Electrical properties such as current-voltage, energy spectrum, and charge collection efficiency was measured and calculated, and compared to previous published works. The experiment results are leading-edge. The discussion however, is not thorough.
Here are some suggestions for the authors which may help to improve the quality of the manuscript:
Line 15 and 19, abbreviations of SBDs and DL is not defined. Figure 1, it is better to show the composition of the ohmic contact and mark the thickness of each layer, so details geometric parameters are shown. Figure 1. At the end of the figure title, there should be a period, correct other figure titles as well. Table I, how the SIMS results of 7.6x10^14 was calculated? If it is Nd-Na, then wouldn’t it be 154.79-147.09=7.79? Please clarify. Line 47, There is another paper you may want to cite: Padhraic Mulligan, et al., Evaluation of freestanding GaN as an alpha and neutron detector, NIMA 2013. This is one of the few results of the sandwich structure detector. Line 92, the first line hanging is different throughout the manuscript, please make is consistent. Line 110, there is a colon after the subtitle. Line 130, ‘CCE is the ratio of the absorbed energy and incident energy.’ Then in line 134 and 161, you are using detected energy. If they are meaning the same thing, please make them consistent. Line 162, ‘reduced DW decreases the path length of the alpha particle.’ The reduced DW cannot change the alpha particle penetration length in GaN, it is the path inside the depletion region that reduced, please clarify. Line 167, ‘The low variation in CCE is mainly due to the presence of thick DW even at low bias conditions, which was achieved by reduced CCD by compensation of DL.’ I don’t think the depletion width is large at low bias voltage of 20 V based on the doping level of 7.6e14. You can theoretically calculate the DW vs. voltage based on the doping level of 7.6E14. Regarding to this, one suggestion is to add a figure on top of Figure 2: use energy as bottom x-axis, voltage as top x-axis, and thickness as y-axis. Then you can clearly show the bias voltage dependent depletion width. If the low CCE variation is not due to the large variation of DW, then it may due to the fact that the e-h pairs generated in the region other than the depletion region (named diffusion region, you can take a look at this article: J Wang, Transient current analysis of a GaN radiation detector by TCAD, NIMA, 2014 ) can also be collected due to the high purity of your GaN, thus less recombination is occurred. Line 193, (see figure 8), it should be figure 10. Line 195, ‘This increase in FWHM is due to spectral widening when the source is placed at an angle to the detector resulting in a reduction of spectral resolution.’ What do you mean by ‘spectrum widening’, please clarify. I guess there is another mechanism, when the incident alpha particle is angled, the alpha particle incident at the periphery region of the Schottky contact can only have partial of its track within the depletion region, and the variation of the track length within the depletion region, which depends on the interaction location relative the center of the Schottky diode that causes broaden of the energy spectrum. Reference 6, the names of the first two authors are not correct. Reference 10, the reference has different style. Please also check and follow the Journal’s recommended style. Reference 15, you can add or replace this reference to another one:J Wang, et al., Review of using gallium nitride for ionizing radiation detection, Applied Physics Reviews 2, 031102 (2015).
Round 2
Reviewer 2 Report
After the revision, and in general, the manuscript was improved but the additional featured regarding capacitance-voltage data should be carefully explained, as, in my opinion, have some issues. Additionally, some small things requires attention. The authors need to consider the following points:
a) Change the reference [30] for a more (and general) suitable reference, for instance some article. The question is related to the fact that such equation (the simple geometric capacitance) has restrictions to be fully applied;
b) Related with such capacitance measurements (figure 7), and compared with the current-voltage data (figure 6), how the authors explain the fact that immediately after 0V and in forward bias we see an electrical carrier injection but the capacitance still the same without any particular change. We expect an increase up to the built-in voltage;
c) By the way, why the authors don’t have the I-V data extended up to 5V in forward bias? And have any idea about the Vbi (built-in voltage)?
d) Check again the grammar and small English things. For instance, line 141 starts with “2%An uniform DW of ~15 μm was measured at all voltages…”
